# Coping strategies of Ghanaian couples after unsuccessful infertility treatment: An exploratory qualitative study

Stephen Mensah Arhin[1]*, Kwesi Boadu Mensah[2], Isaac Tabiri Henneh[3], Felix Yirdong[4,5], Evans Kofi Agbeno[6], Charles Ansah[2], Martins Ekor[1]

1 Department of Pharmacology, School of Medical Sciences, University of Cape Coast, Cape Coast, Ghana, 2 Department of Pharmacology, Kwame Nkrumah University of Science and Technology, Kumasi, Ghana, 3 Department of Pharmacotherapeutics and Pharmacy Practice, School of Pharmacy, University of Cape Coast, Cape Coast, Ghana, 4 Department of Psychological Medicine and Mental Health, School of Medical Sciences, University of Cape Coast, Cape Coast, Ghana, 5 Department of Psychology, The Graduate Center, City University of New York, New York, United States of America, 6 Department of Obstetrics and Gynecology, School of Medical Sciences, University of Cape Coast, Cape Coast, Ghana

* stephen.arhin@ucc.edu.gh

## Abstract

### Background

Psychological distress and social burdens associated with infertility among couples have been well-documented. However, little is known about the specific coping strategies employed by couples in low-middle-income countries such as Ghana, in the aftermath of unsuccessful infertility treatment attempts. In this qualitative study, we explored specific coping strategies patients adopt to address psychological distress related to unsuccessful treatment for infertility.

### Methods

A semi-structured interview approach was used to elicit qualitative responses from 18 fertility clients after unsuccessful treatment at four fertility clinics in Ghana. Thematic analysis (TA) was used to examine the coping strategies adopted by participants in response to psychological distress associated with infertility treatment failures. This allowed us to explore potential culturally specific coping strategies employed by participants in response to infertility-related psychological distress.

### Results

The themes that emerged as coping strategies in response to infertility-related psychological distress were diversional activities, intrapersonal cognitive reframing, social isolation, familial support, religious coping, avoidance-focused coping strategies, seeking encouragement, and professional help.

**Data availability statement:** All relevant data are available in the Supporting Information file.

**Funding:** The author(s) received no specific funding for this work.

**Competing interests:** The authors have declared that no competing interests exist.

## Conclusion

The findings from this study indicate that coping strategies that involve isolating oneself may not provide lasting emotional relief for individuals experiencing infertility. Relational activities contribute positively to coping. This is relevant in helping health professionals in the management of infertility treatment failures, which may include setting up support groups of similar experiences to draw strength from each. Furthermore, the results underscore the need to integrate psychological interventions into the counseling of couples following an unsuccessful infertility treatment. The clinical and research implications of these findings are discussed.

## Introduction

Infertility- a major public health concern, is a significant source of emotional and psychological stress for couples worldwide [1–4]. In pronatal societies such as Ghana, where cultural norms strongly emphasize childbearing as a non-negotiable outcome after marriage and a benchmark for successful family life, unsuccessful infertility treatments can result in significant emotional distress and heightened social pressure [5]. For instance, psychological distress, including depression, anxiety, and stress, has been reported frequently among couples following an unsuccessful infertility treatment [2]. These societal expectations about childbearing in Ghana not only impose an additional layer of stress on the couples but may also influence the coping strategies they deploy to address psychological distress following unsuccessful infertility treatment [2,6]. Nonetheless, research is yet to explore the coping strategies adopted by Ghanaian couples to manage psychological distress related to unsuccessful infertility treatment to inform the provision of holistic care.

The longstanding assessment and treatment of infertility does not always yield the desired results as expected [3], which compels couples to adopt various coping strategies. The stress of treatment may compound psychological distress, which can influence the decision to discontinue seeking medical care [6], and this emphasizes the need for stringent coping mechanisms. The coping strategies adopted by couples following unsuccessful infertility treatment may vary, depending on their cultural background, social support systems, and personal resilience [6–8].

Lazarus and Folkman's 1984 stress and coping theory proposed two types of coping strategies – problem-focused and emotion-focused coping strategies, in response to distress. They conceptualized coping as the process by which a person appraises an internal or external demand as taxing and adjusts their cognitive and behavioral resources to manage the perceived demand [9]. They further posited that individuals may use either problem-focused and/or emotion-focused coping strategies to manage stressors [9]. Problem-focused coping – is related to all cognitive and behavioral efforts to directly address the stressful situation and potentially alter an environment

or eliminate the source of stress. Whereas emotion-focus in coping cognitive and behavioral effort is directed at regulation or reducing emotional response to perceived stressors [9,10].

Emotion-focused coping, such as seeking spiritual comfort through prayers, fasting, and reliance on religious leaders, is common and reflects the generality of religiosity in Ghanaian society [5,11]. Moreover, couples may turn to social support from family, friends, or support groups as a means of alleviating their emotional distress and regaining a sense of normalcy [1]. However, it is important to note that not all coping strategies adopted in maintaining a balance yield positive results. Repressive or avoidant coping mechanisms, such as denial and withdrawal, are also prevalent among couples experiencing the stigma of infertility following unsuccessful treatment attempts [12]. For example, women may internalize blame and experience greater psychological distress, as societal norms often hold them accountable for infertility [13]. On the other hand, some couples may adopt problem-focused coping strategies, such as seeking alternative treatments as a way of regaining control over their circumstances [11]. However, little is known about the coping strategies adopted by Ghanaian couples seeking treatment for infertility, which may often be influenced and shaped by deep-rooted cultural and religious beliefs [14].

Prior studies have found that different coping strategies may yield different psychological outcomes among couples after an unsuccessful infertility treatment. For instance, repressive or avoidant coping mechanisms, such as denial and withdrawal, are also prevalent among couples experiencing the stigma of infertility following unsuccessful treatment attempts [12]. Gender differences in psychological distress related to (unsuccessful) infertility treatment have also been reported elsewhere, with women reporting disproportionately affected and using different coping strategies compared to men/husbands [15]. Coping strategies such as seeking social support, escape-avoidance, planful problem-solving, and positive reappraisal were reported among women in response to infertility-related distress [15]. Furthermore, some couples may adopt problem-focused coping strategies such as seeking alternative treatments to gain a sense of control over their circumstances [11,16].

Currently, the treatment protocol for infertility in Ghana prioritizes pharmacotherapy as the primary intervention [17], largely due to resource constraints [18], and the lack of financial support [6].

Understanding the coping strategies employed by Ghanaian couples following unsuccessful infertility treatment is essential for developing culturally sensitive and targeted interventions that improve emotional and psychological well-being, contributing to their holistic care. By clarifying the intersection of culture, religion, and coping mechanisms, healthcare providers and policymakers can provide tailored support to couples experiencing emotional and psychological distress related to infertility. Therefore, the current study seeks to qualitatively explore the coping strategies employed by couples after unsuccessful infertility treatment.

## Materials and methods

### Participant selection

The present study was embedded within a broader investigation that sought to evaluate the effectiveness of pharmacological interventions among infertile couples seeking fertility care.

They were monitored over a 12-month period to track pregnancy outcomes. The patients were given special identification numbers during their initial visit to the facility for assessment and evaluation. Details regarding their sociodemographic data were obtained after consultation with the doctor. The patients were asked to report to the facility for review after they had completed their prescribed treatments. Those who were unable to conceive, and voluntarily agreed to be part of the study were engaged and interviewed after meeting the inclusion criteria. The semi-structured interviews were carried out over weeks during follow-up visits to the facility. The study was conducted in four selected fertility clinics within the Cape Coast Metropolis in the Central Region of Ghana. The selected facilities have specialist gynecologists and urologists who attend to most fertility cases within the metropolis. A flowchart showing the selection of respondents for interviews is shown in Fig 1.

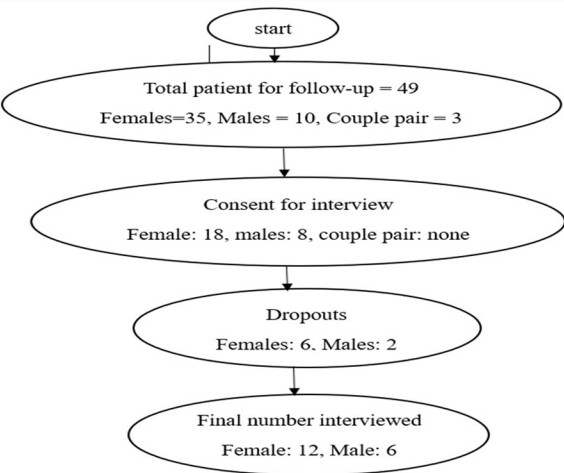

**Fig 1. A flowchart showing the selection of respondents for interview.**

## Conducting the interviews

Face-to-face interviews were conducted in a special room assigned to only the principal investigator and the patient. The interviews were conducted for individual partners who were available during the follow-up visits. Each interview lasted between 15 and 20 minutes. Permission was sought from individual participants before the interviews were audio-recorded. To ensure accuracy and effectiveness in communication, interviews were conducted in the preferred spoken language of participants' — (English, and Twi-a local dialect)—between March and June 2021. Sample questions that were asked during the interview include the following: *How do you cope with the emotional challenges of infertility? How do you manage the uncertainty: and unpredictability of infertility so far? Can you describe a difficult decision you've had to make regarding infertility?* The interviews that were conducted in *Twi* were transcribed verbatim by the principal investigators. To ensure that the meaning was not lost, three of the investigators who were familiar with the local language were allowed to listen to the audio-recordings several times before a consensus was reached. This was done to preserve the original meaning of participants responses. Rigorous qualitative strategies were employed to ensure trustworthiness in this study as defined by Lincoln & Guba [19]. For instance, interviews and engagements with participants were prolonged and research team members deliberated on emergent meanings of teams to ensure *credibility* of our themes. Also, examined participants responses to provide thick and descriptive accounts of their experiences following unsuccessful fertility treatment – this was to ensure transferability as a quality check for our study. Throughout this iterative and reflexive process, we maintained detailed documentation of the analytic process, journaled our reflexive process, and iterative coding to ensure transparency and address our potential biases. This ensured that themes from our responses and meanings were *dependable*. As a quality check – *confirmability* was enhanced through bracketing and peer debriefing to reduce interpretive bias and foreground participants' authentic voices. Together, this ensured that trustworthiness of our qualitative process and the meanings from generated themes.

## Data analysis

Using descriptive statistics, participants' sociodemographic characteristics were examined. Qualitative data was analyzed manually using Thematic Analysis (TA). TA enables the researcher to group the entire textural data into a list of common themes as a true reflection of the data [20]. The audio-recorded interviews were transcribed verbatim, and each transcript

was read several times by the researchers to understand the true meaning of the conversation. The key ideas in the form of phrases, sentences, or paragraphs emerging were assigned codes, and the codes were sorted into themes based on their similarities. Each of the three members of the research team coded the data independently and a series of meetings were held before agreeing on common themes and subthemes.

Moreso, TA was considered the most appropriate analytical tool for this study considering the researchers' active involvement in the process and the level of thoughtfulness required to systematically identify codes, analyze, develop themes, and interpret patterns within the interview responses [20]. This allows for an in-depth understanding of couples' experiences while exploring their coping strategies following unsuccessful infertility treatments.

The TA process was guided by the steps proposed by Braun and Clarke, which involved; i) a data familiarization phase; ii) generating initial codes; iii) generating and developing and reviewing initial themes; iv) defining and naming themes; v) and producing a comprehensive final report. These steps involve rigorous approaches that seek to accurately capture the semantic and latent meanings of participants' responses to experiences and coping strategies following unsuccessful infertility treatments [20].

### Ethical consideration

Ethical clearance for the study was obtained from the Institutional Review Board of the University of Cape Coast (ref No: UCCIRB/EXT/2019). An informed written consent was also obtained from individual participants before the interviews were conducted. The participants were informed that participation was voluntary. Given the sensitive nature of the study, extra precautions were taken to protect participants' confidentiality. This included assigning special codes to mask identifiable details, and securing collected data in a password protected computer.

## Results

### Sociodemographic characteristics of the respondents

Table 1 shows the sociodemographic data of the participants. The mean age of all participants in the study was 35±5.45. The duration of infertility among the participants ranged from 2–10 years. There were diverse occupations among the respondents, with teaching being the predominant profession.

### Coping strategies of the participants

Three main areas of coping strategies were assessed: with themes and subthemes emerging from each section. The three key areas assessed were emotional coping strategies; coping with uncertainty; self-identity and self-esteem; and decision-making and coping with setbacks. Detailed information regarding the summary of themes and sub-themes for each section is illustrated in Table 2.

### Emotional coping strategies

Concerning the emotional coping strategies of participants, various themes emerged. Among them include diversional activities, divine support, family support, social disengagement, and self-motivation.

**Diversional activities.**  Diversional activities were among the key emotional coping strategies employed by the participants. This is very key regarding avoiding stressors that trigger emotional disturbances due to infertility. A female respondent (FR-4) recounted her story as follows:

*I have been going through a lot since I started experiencing this problem. Sometimes I don't even know what to do. I become emotionally down and unable to focus on certain things. I try to relax and sometimes find a rest by sleeping if I am at home.*

**Table 1. Socio-demographic data of respondents.**

| Variables | Occupation | Duration of infertility (yrs) |
|---|---|---|
| Age±SD (yrs) of respondents = 35±5.45 | | |
| Coding Characteristics | | |
| Female respondents | | |
| FR-1 | Teaching | 4 |
| FR-2 | Trading | 6 |
| FR-3 | Lab technician | 5 |
| FR-4 | Teaching | 4 |
| FR-5 | Seamstress | 9 |
| FR-6 | Sales representative | 2 |
| FR-7 | Schooling | 7 |
| FR-8 | Farming | 8 |
| FR-9 | Teaching | 3 |
| FR-10 | Fish monger | 4 |
| FR-11 | Teaching | 2 |
| FR-12 | Farming | 10 |
| Male respondents | | |
| MR-13 | Commercial driver | 5 |
| MR-14 | Teaching | 4 |
| MR-15 | Auto mechanic | 3 |
| MR-16 | Farming | 6 |
| MR-17 | Health worker | 5 |
| MR-18 | Accountant | 8 |

Other study participants recounted using similar emotional coping strategies that enabled them to deal with stressors associated with infertility. Such emotional coping techniques are key to reducing anxiety and helplessness following a series of unsuccessful treatments. Some of such strategies were recounted by a female respondent (FR-5) as follows:

*Anytime I think of the problem I am going through; my heart becomes disturbed. I tend to lose focus on whatever I am doing…. Is like it has taken over all my concentration. I sometimes engage in a conversation with a close friend just to take my mind off some stuff. I do video calls with closed family relations, watch television and sleep.*

A male participant also reported similar experience by sharing the techniques below:

*Sometimes I leave home to engage myself in social activities such as watching football with friends. At other times too, I engage in hiking or cycling just to keep myself busy so I won't continue thinking about the problem (MR-13).*

**Divine support.** Divine support was found to be one of the major emotional coping strategies used by the participants. Both male and female respondents recounted their faith in God as a source of hope for childbearing.
For instance, one female participant (FR-3) said:

*I always pray to God for his mercy upon my life so that I can have another child. He is the only source of our hope. My pastor has also been praying with us for God's grace to conceive*

Another female respondent (FR-5) also expressed her faith in God in this manner:

**Table 2. Themes and sub-themes that emerged from the interview.**

| Categories | Number | Themes | Sub-themes |
|---|---|---|---|
| **1.Emotional Coping Strategy** | 1. | Diversional activities | 1. Takes mind off the problem |
| | | | 2. Concentrates on work |
| | | | 3. Further education |
| | | | 4. Converse with friends |
| | 2. | Divine support | 1. Prays to God |
| | | | 2. Hopes in God |
| | | | 3. Believes God for solution |
| | 3. | Family support | 1. Parental assurance |
| | | | 2. Siblings' assurance |
| | 4. | Social disengagement | 1. Avoids triggering event |
| | | | 2. Isolation from friends and colleagues |
| | 5. | Self-motivation | 1. Self-assurance |
| | | | 2. Hopes in the future |
| **2.Coping with Uncertainty** | 1. | Repression | 1. Takes mind off the problem |
| | | | 2. Engages positive thinking |
| | | | 3. Remain focused |
| | | | 4. Keep self-busy |
| | 2. | Seeks advice | 1. Seeks professional help |
| | 3. | Divine support | 1. Pursue medical treatment |
| | | | 2. Hopes in God |
| | 4 | Reassurance | 1. Keeps praying |
| | | | 2. Self-assurance |
| | | | 3. Family assurance |
| **3.Decision making and coping with setbacks** | 1. | Seeks support | 1. Seeks professional treatment |
| | | | 2. Hopes in positive treatment outcome |
| | 2 | Seeks encouragement | 1. Self-assurance |
| | | | 2. Parental encouragement |
| | | | 3. Spousal encouragement |
| | 3 | Avoidance | 1. Withdrawal from treatment |

*I will not give up yet. I only pray for more of Gods strength and count on him for a better outcome. With God all things are possible.*

A male participant also recounted his faith in God as a form of coping emotionally with the distress associated with infertility when he said:

*Prayer is my motivation. I have the strongest hope in God that I will surely hold my baby one day. God will not disappoint us (MR-14).*

**Family support.** Family support also played a key role in ensuring the emotional well-being of the. Although some families may compound the psychological distress of infertile patients due to delayed childbearing, others also serve as a major source of hope through which individuals struggling with infertility and having undergone a series of treatments without success rely on emotional support. Support from the family plays a significant role in alleviating the emotional

burden associated with infertility, especially in a typical pro-natal society where most families put undue pressure on their relatives because of infertility. One participant had this to say concerning family support:

*Anytime I become sad about the situation, I call my mother. She has been reassuring me. She encourages me all the time and that gives me hope. Sometimes my parents even suggest some places to go for treatment and support me financially too (FR-10).*

**Self-motivation.** Self-motivation requires the use of an inner drive to avoid emotional stressors of infertility. This provides a sense of direction that are within one's control. For instance, a male respondent (MR-17) made a statement on self-motivation as a coping strategy for dealing with emotional stress associated with infertility when she said:

*That positive pregnancy even came when we were least expecting anything like that. Although it didn't stay due to mis-carriage, it gives us assurance that she will definitely conceive again.*

A similar statement was made by FR-8:

*Friends are expecting that you conceive, church members, as well as family members. At times, when they ask you about it, they may sound like they are reassuring you, but that keeps reminding me you the emotional distress. I only reassure myself and call friends to converse with.*

Another respondent recounted:

*It makes me lose focus sometimes. But always reassure myself that, once we are alive, there is still hope for the future (FR-12).*

**Social disengagement.** The pressure from friends, families, and society during times of infertility can be traumatic. Distancing oneself can be a form of protective barrier by offering some breaks from emotionally disturbing scenes. One participant (FR-11) recounted this strategy when she said:

*Is not something that you can easily develop antidote to it and say you have forgotten about it. But in all these, isolation from certain places decreases the anxiety.*

A male participant also disclosed his emotional coping strategy in the form of social disengagement by saying:

*Although at times, and in some gatherings, some friends may intentionally pass comments that depicts that if I can't work on the woman, somebody will take her from me, all because she is not getting pregnant, but I just ignore them, and sometimes leave the scene to avoid further emotional agony (MR-13).*

## Coping with uncertainty

Uncertainty characterizes various aspects of infertility such as making plans, outcomes of investigations, and treatment that can compound the psychological distress of these infertile patients. Therefore, identifying how infertile patients cope with uncertainty is central to mitigating the psychological trauma that these patients experience. The following are some strategies that infertile patients use when dealing with uncertainty.

**Repression.** One of the major ways in which infertile patients cope with uncertainty is through repression. This coping mechanism may provide immediate and temporary relief for patients struggling with infertility, as it enables them to block unfavorable thoughts and memories from awareness.

One female participant (FR-6) shared her experience when she said:

*If you decide to always think about when you are going to get outcome, you will always feel sick. So, the best thing is to take your mind off for some time and focus on what you are doing.*

Another participant stressed the need to remain focused as she recounted:

*I try as much as possible to erase it from my thoughts and everything that I am doing. It is not all that easy but that is what helps me to get going with other businesses for survival (FR-9).*

Similarly, FR-10 reaffirmed remaining focused as a way of coping with infertility when she said:

*I try to remain focused on what I am doing because if I decide to think about it all the time, I cannot even do my work well*

A male participant (MR-14) had this to say:

*You only need to take your mind off it and that is all. If you decide to think about it every day, you may even die early.*

**Seeks advice.** Infertile patients cope with the problem by seeking advice from health professionals that enables them to deal with life stressors. One respondent (FR-12) shared her view on this by saying:

*I reassure myself that once we are seeking professional help, we will get the results we need. The doctors are doing all that they could, and we hope for the best.*

Similarly, a male participant (MR-15) shared his views as he disclosed:

*I also take consolation from the fact that it is not over yet. Even if we need to go for assisted reproduction, we will do so. We can't give up at this stage*

**Reassurance.** Reassurance is another coping mechanism employed by respondents in dealing with the uncertainties associated with infertility by offering a sense of hope and positivity.
One female participant (FR-2) described using reassurance as a way of dealing with uncertainties due to infertility:

*I always have to get closer to my parents in order to become okay. My parents and my siblings are my source of joy whenever I see them*

One participant also described reassuring herself when she said:

*Sometimes I give up, although I believe that it is not all over yet since this is just the beginning of treatment. I listen to some people's success story and I say to myself, it is all not over yet (FR-4).*

Parental reassurance was also disclosed by FR-8 as follows:

*When I am down emotionally because of these thoughts, I call my parents and they reassure me.*

## Decision making and coping with setbacks

Making decisions and dealing with setbacks during the time of infertility can be frustrating and marked by periods of uncertainty about the future. Patients employ a variety of strategies to manage such emotional aspects of infertility. The coping strategies used by infertile couples during such emotional dilemmas include the following:

**Seeks support.** Support-seeking is a crucial means of coping with infertility setbacks, as it helps individuals and couples navigate the emotional and psychological challenges of the process. Such support includes seeking professional advice for the next line of action when previous treatment attempts fail. One female participant described her experience by saying:

*I just hope that my next available treatment will bring me positive results. We have trust in the professionals and I know they are doing their best to help us. We will continue to seek their advice on the next alternative option (FR-8).*

A male participant (MR-14) also disclosed that she hopes for a positive treatment outcome despite the current outcome when he said:

*I don't want to be easily moved by these adverts although is not easy to do that. I only try as much as possible to rely on what health professionals are doing for me, and hope for a better outcome until they say something different.*

**Encouragement.** Individuals facing decision-making challenges and setbacks due to infertility resort to various forms of encouragement as a way of coping with such stressors, as such encouragement provides hope, resilience, and strength in the face of difficulty in childbearing.

A female participant (FR-4) recounted seeking parental advice and encouragement when she said:

*I try to discuss the problem with my mother whenever I encounter new challenges or setbacks. I am so close to my mum, and she sometimes offer me the help and encouragement that I need in any difficult situation.*

A male participant (MR-16) described how he sought advice and encouragement from the spouse, and his parents as he disclosed:

*When it comes to decision taking and issues with setbacks concerning the problem, especially about seeking treatment and where, and the outcome, I discuss with my partner. We usually have no challenge about where and how to seek treatment. Only that whenever I suggest things like local herbals, my partner usually disagrees and that can hurt a bit. Knowing the kind of testimonies people give concerning such treatments. But in all these, I seek advice from my mother on what to do next, especially when previous treatment didn't go well as expected*

Another participant (FR-5) recounted using self-assurance to cope with difficult situations and setbacks as she disclosed:

*I was hoping to get the expected results but it didn't go that way. That is not easy to deal with, but since that is not the end of it all, I only assure myself that I will conceive one day*

Also, MR-18 recounted his reliance on family, faith, and inner drive as a source of encouragement by saying:

*We also seek family support too. They also offer a lot of advice and motivation. Our pastor too has been praying for us. I also try as much as possible to concentrate on my job and not to focus on only the problem*

## Discussion

This study explored the coping strategies of infertile patients after unsuccessful treatment for infertility in four fertility clinics in the Central Region of Ghana. The study revealed various forms of coping strategies adopted by infertile patients to address infertility stressors. The areas of coping that were assessed were emotional coping strategies, coping with uncertainty, and coping with decision-making and setbacks.

Coping with infertility can be a complex process that encompasses a variety of approaches and strategies. Such coping strategies after unsuccessful infertility treatment often involve engaging in diverse activities. These activities are adopted to alleviate emotional distress and foster a sense of normality and relief [21]. Various coping strategies such as problem-focused, and emotion-focused coping play important roles in lowering levels of psychological distress and feelings of isolation [22,23].

An important aspect of coping that was assessed in the current study was the emotional coping strategies adopted by the participants. Couples struggling with infertility undoubtedly face all sorts of emotional distress that may interfere with daily functioning [2]. Adopting strategies to address such stressors is critical in maintaining the overall health of the individual [23]. Coping with the emotional toll of unsuccessful infertility treatments usually require strategies that underscore social and emotional resilience. Among these, seeking *divine support, family support, social disengagement,* and *self-motivation* emerged as prominent mechanisms adopted by patients. Each of these mechanisms is essential in providing a unique pathway for individuals to process their experiences and foster optimism [1,21,24].

One of the major emotional coping strategies adopted by individuals during stressful periods of infertility was *divine support*. This strategy allows individual patients to turn to religious or spiritual practices as a way of finding solace and meaning in their journey [24]. Spiritual beliefs, according to Latifnejad et al, serve as a foundation for resilience, providing a sense of purpose and helping individuals manage the uncertainties associated with infertility [25]. Another important strategy in the form of family support serves as an emotional anchor, offering empathy, encouragement, and practical assistance during the period of unsuccessful infertility treatment. Open communication with family members fosters a sense of belonging and reduces the feelings of isolation commonly experienced by infertile couples [21,26]. Coupled with family support, most patients also adopted self-motivation as a way of fostering internal resilience and stability. Self-assurance, hope in the future, practicing positive self-talk, and engaging in activities that enhance self-worth are ways of developing inner goals and ensuring stability in daily routines. According to Moutzouri et al, individuals who focus on self-improvement and maintain a proactive mindset often report higher levels of emotional well-being [27]. This approach enables couples to regain a sense of control over their lives, despite the challenges posed by infertility.

In contrast, some individuals adopt *social disengagement* as a means of coping. This was achieved through limiting interactions with others, especially in situations that might trigger emotional distress. Such avoidances create a protective emotional buffer by allowing individuals to process their emotions privately and avoid unnecessary stressors [28]. Although important in ensuring emotional stability in the short term, long-term social distancing may exacerbate loneliness; therefore, individuals who adopt such strategies must be educated to do so with caution.

Another important area of coping that was assessed after unsuccessful infertility treatment was how the patients cope with uncertainty. Coping with uncertainty is a significant challenge for infertile couples in Ghana following unsuccessful treatment. Anxiety due to uncertainty about possible delays in treatment outcomes poses significant psychological distress, survival, and daily functioning [7,29,30]. As a result, most patients develop diverse coping mechanisms to enable them to navigate through life. Some of the key coping strategies that patients adopt while dealing with uncertainty include repression, advice seeking, and reassurance, similar to other study reports [31].

One prevalent coping mechanism in dealing with uncertainty is *repression*. This involves individuals suppressing their emotions and avoiding discussing their struggles openly with other people. This strategy is often influenced in part by societal stigma surrounding infertility, which discourages public expression of pain or frustration [32]. While repression may

 

provide temporary relief, it can lead to unresolved emotional burdens over time. Studies have shown that individuals who repress their emotions often experience heightened psychological distress and feelings of isolation [33].

Another important approach is *seeking advice* from trusted individuals and healthcare professionals. For example, in Ghana, advice from respected figures is highly valued and often serves as a guiding force for coping. Annan-Frey et al reported that couples who seek advice from experienced individuals and professional sources report a greater sense of direction and hope, as they bury unwanted thoughts [23].

In a similar way, infertile patients often seek *reassurance* to cope with uncertainty. Reassurance may come from family, self, and spiritual leaders, affirming that their struggles are not unique and that solutions may eventually be found. According to Donkor et al, reassurance is particularly vital in Ghanaian communities, where collective support plays a substantial role in fostering emotional resilience [5]. Reassurance can mitigate feelings of despair by reinforcing hope and the importance of patience.

Moreover, decision-making and the ability to cope with setbacks after unsuccessful infertility treatments present significant challenges for couples in Ghana. The sociocultural setting influences how couples respond to these experiences, with strategies such as seeking encouragement and avoidance being prominent.

One common approach is seeking encouragement from friends, family members, and spiritual leaders, as such mechanisms help the patients navigate the uncertainty of infertility. Encouragement provides a sense of belonging, which eventually reduces feelings of isolation and despair. The communal nature of societies in Ghana implies that support networks play a central role in decision-making processes. Individuals who actively seek encouragement are more likely to view their situation with hope and optimism. This enables them to consider alternative options such as adoption or further medical interventions [34,35].

On the other hand, avoidance is another coping mechanism frequently adopted by infertile patients during the period of decision-making and setbacks. This strategy involves withdrawing from social interactions or avoiding discussions related to infertility. Arhin et al noted that avoidance is often a response to societal stigma, which can aggravate feelings of shame and insufficiency [6]. While avoidance can provide temporary relief by shielding individuals from stress-inducing situations, it may hinder long-term decision-making and the pursuit of alternative solutions. However, prolonged avoidance can result in emotional disconnection and tension in relationships, particularly if partners or extended family members feel excluded from the decision-making process.

While much of the existing discourse on coping strategies in the current study has focused on the individual emotional toll, viewing coping dynamics as a shared experience within the couple dyad could be of prime importance. Couples often navigate treatment decisions, emotional distress, and social pressures together—yet their coping mechanisms may diverge or align in complex ways. These relational patterns indicate that infertility does not merely impact individuals in isolation but reshapes relationship dynamics, gender roles, and expectations [36]. Couples who actively engage in joint problem-solving or shared spiritual and emotional resources appear to cope more adaptively, underscoring the value of couple-based interventions [7,36].

To foster deeper resilience and effective interventions, infertility care in low-resource settings should integrate relational counseling and partner-inclusive support services. Such efforts can help normalize shared emotional processing, reduce the potential for relational strain, and empower couples to approach treatment and coping as a united front [7,36].

The outcome of the current study suggests that many couples seeking fertility care may experience grief, isolation, and social stigma after treatment failure, and may resort to various coping strategies, some of which may offer only temporary or limited relief. This therefore requires that health systems mandate psychosocial counseling services as part of fertility care — not just pre-treatment but also post-treatment. This is important in low-resource settings, especially in sub-Saharan Africa where infertility is both common and under-addressed due to cost, stigma, and policy neglect. Addressing infertility holistically may improve quality of care and reduce emotional trauma, especially in societies where parenthood is highly valued.

The major limitation of the study was the low attendance rate of male respondents during the review period, which made it difficult to get a sizeable number of men respondent to the interviews. Additionally, the unavailability of both partners during the review visit made it impossible to interview the couple pair. The findings may disproportionately reflect women's experiences, coping strategies, and viewpoints, potentially overlooking how men experience and respond to infertility. This may lead to gender-biased conclusions, portraying infertility as a predominantly female burden, when in fact it affects couples almost equally. These may also limit insight into shared coping strategies or communication dynamics between couples. Additionally, the limited number of male participants in this study presents a challenge to the generalization of its results. Since the data largely represent the viewpoints of women, the conclusions drawn may not adequately encompass the gender-related aspects of dealing with infertility. As a result, the relevance of these findings to wider or more varied groups—especially those in which male participation in reproductive health is significant—might be constrained. Future studies may consider developing male-friendly recruitment strategies such as male-focused support groups, and allowing more time to engage male participants, or offering incentives to encourage male participation, especially in sub-Saharan region where stigma or norms limit male involvement. However, the interviewing of the respondents individually made it possible to examine the peculiar coping mechanisms adopted by individual partners. Additionally, the study did not formally incorporate mechanisms for emotional support or referrals to counseling services. Given the sensitive nature of infertility, we recommend that future studies integrate structured emotional support or referral systems as part of ethical safeguards.

## Conclusion

The findings from this study indicate that coping strategies that involve isolating oneself may not provide lasting emotional relief for individuals experiencing infertility. Relational activities contribute positively to coping. This is relevant in helping health professionals manage couples with infertility treatment failures, which may include setting up support groups of similar experiences to draw strength from each other to make their challenge more bearable. Furthermore, the results underscore the need to integrate psychological interventions into the counseling of couples following an unsuccessful infertility treatment. Health professionals, counselors, and community leaders can play a significant role in guidance as a way of promoting healthier coping. Additionally, to promote holistic reproductive healthcare in low-middle income settings, national health policies must prioritize the emotional and social dimensions of infertility.

## Supporting information

**S1 File.  Interview transcripts.**
(DOCX)

## Acknowledgments

The authors wish to express their sincere gratitude to all the respondents who were available for this study. We also wish to thank the management of the various fertility centers for making their clinics accessible for this research.

## Author contributions

**Conceptualization:** Stephen Mensah Arhin, Kwesi Boadu Mensah.

**Data curation:** Evans Kofi Agbeno.

**Formal analysis:** Stephen Mensah Arhin, Isaac Tabiri Henneh, Felix Yirdong.

**Funding acquisition:** Stephen Mensah Arhin, Kwesi Boadu Mensah.

**Investigation:** Stephen Mensah Arhin, Isaac Tabiri Henneh, Evans Kofi Agbeno.

**Methodology:** Stephen Mensah Arhin, Kwesi Boadu Mensah, Isaac Tabiri Henneh, Felix Yirdong, Evans Kofi Agbeno, Charles Ansah.

**Project administration:** Stephen Mensah Arhin, Kwesi Boadu Mensah, Martins Ekor.

**Resources:** Stephen Mensah Arhin, Kwesi Boadu Mensah, Isaac Tabiri Henneh.

**Software:** Isaac Tabiri Henneh, Felix Yirdong.

**Supervision:** Kwesi Boadu Mensah, Charles Ansah, Martins Ekor.

**Validation:** Felix Yirdong, Evans Kofi Agbeno.

**Visualization:** Stephen Mensah Arhin, Isaac Tabiri Henneh, Evans Kofi Agbeno, Charles Ansah.

**Writing – original draft:** Stephen Mensah Arhin, Kwesi Boadu Mensah, Isaac Tabiri Henneh, Felix Yirdong, Charles Ansah, Martins Ekor.

**Writing – review & editing:** Stephen Mensah Arhin, Kwesi Boadu Mensah, Isaac Tabiri Henneh, Felix Yirdong, Evans Kofi Agbeno, Charles Ansah, Martins Ekor.

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
