## [Decision Letter · Decision Letter 0]

PONE-D-25-06994Coping strategies of Ghanaian couples after unsuccessful infertility treatment: An exploratory qualitative study in the Central Region of Ghana.PLOS ONE

Dear Dr. Arhin,

Thank you for submitting your manuscript to PLOS ONE. After careful consideration, we feel that it has merit but does not fully meet PLOS ONE’s publication criteria as it currently stands. Therefore, we invite you to submit a revised version of the manuscript that addresses the points raised during the review process.

We look forward to receiving your revised manuscript.

Kind regards,

Lily Kpobi, Ph.D.

Academic Editor

PLOS ONE

Journal Requirements:

Additional Editor Comments:

The reviewers raise some important points regarding ensuring clarity in the methods and presentation of the findings. It would also be good to expand on the implications of your findings for health policy and practice. How can these fit into the wider context of reproductive health issues in a low income setting?

Reviewers' comments:

Reviewer's Responses to Questions

**Comments to the Author**

1. Is the manuscript technically sound, and do the data support the conclusions?

Reviewer #1: Yes

Reviewer #2: Partly

2. Has the statistical analysis been performed appropriately and rigorously? 

Reviewer #1: N/A

Reviewer #2: Yes

3. Have the authors made all data underlying the findings in their manuscript fully available?

Reviewer #1: Yes

Reviewer #2: Yes

4. Is the manuscript presented in an intelligible fashion and written in standard English?

Reviewer #1: Yes

Reviewer #2: No

5. Review Comments to the Author

Reviewer #1: The article is well written, however there are several important issues that I will like the authors to address.

1. In the abstract section (line 24-25) the authors wrote “A semi-structured interview approach was used to elicit qualitative responses from 18 fertility clients who were seeking treatment for infertility at four fertility clinics in Ghana”. The authors should clarify this because it is unclear whether the study was among those seeking infertility treatment or those who were unsuccessful with their treatment, as indicated in line 19-21 in the abstract session.

2. At the conclusion session of the abstract, the authors wrote “Relational activities helped”. How does it help? The authors should clarify this.

3. In the introduction section (line 53-55), the authors wrote “Nonetheless, research is yet to explore the coping strategies adopted by Ghanaian couples to manage psychological distress related to unsuccessful infertility treatment to inform the provision of holistic care”. I don’t think this is the case. A recent study by Dialo et al., (2024) “Exploring the psycho-social burden of infertility: Perspectives of infertile couples in Cape Coast, Ghana” provided valuable insight into this phenomenon. The current study should be a build-up on the previous one. My understanding is that the current study was conducted in the same setting. So, the authors should highlight coping strategies after unsuccessful treatment and focus on that. This distinction should be made clearer.

4. At the “Materials and Methods” section (Line 130 -131) the authors wrote “Permission was sought from individual respondents before the interviews were audio-recorded”. I suggest the authors change the word “respondent” to “participants” to be consistent with the qualitative method of inquiry adopted by the authors. Please, go through the manuscript and correct this accordingly.

5. With regards to conducting the interviews, did the interviewer interview the participants as couples or individually? If individually, how were the female participants matched with the male participants? If they were interviewed as couples, what is the justification? The authors need to clarify these issues. My understanding from the limitation session is that the interviews were conducted individually. So, this needs to be made clear in the write-up.

6. Line 136-137, the authors wrote “The interviews that were conducted in Twi were transcribed verbatim by the principal investigators”. What mechanisms were put in place to ensure that meaning is not lost during the transcription and translation processes? The authors need to provide readers with that information.

7. With regards to the data analyses, did the authors employ any qualitative data analyses software? My understanding is that the analyses were done manually. This should also be made clear in the write-up.

8. In table 2: Themes and sub-themes that emerged from the interview, Category: 2 Coping with Uncertainty, the sub-theme under “Divine Support” “2. Pursue treatment”. Can this sub-theme be clarified? What kind of treatment? Spiritual treatment? Spiritual-herbal treatment??? Or what? The authors need to clarify this statement.

9. Some of the quotes do not have unique identifiers. The authors should read through the entire manuscript and ensure that all the quotes have unique IDs.

10. Can the authors replace the concept “social distancing” with “self-imposed social isolation” or “social disengagement”? Social distancing in this context could be misleading.

Reviewer #2: The methodology seems well-structured and relevant for the research objectives, particularly in addressing emotional, psychological, and social coping mechanisms. However, refining data collection techniques, and enhancing discussion of data validity and ethical considerations could strengthen the study’s rigor and impact. Further comments in the manuscript.

6. PLOS authors have the option to publish the peer review history of their article (what does this mean? ). If published, this will include your full peer review and any attached files.

**Do you want your identity to be public for this peer review?** For information about this choice, including consent withdrawal, please see our Privacy Policy .

Reviewer #1: No

Reviewer #2: No

---

## [Author Response · Author response to Decision Letter 1]

14 May 2025

Response to reviewer’s comments

Journal’s requirements

Comment: When submitting your revision, we need you to address these additional requirements. Please ensure that your manuscript meets PLOS ONE's style requirements, including those for file naming

Response: The manuscript has been revised to meet the journal’s requirements

Comment: Please provide additional details regarding participant consent. In the ethics statement in the Methods and online submission information, please ensure that you have specified (1) whether consent was informed and (2) what type you obtained (for instance, written or verbal, and if verbal, how it was documented and witnessed). If your study included minors, state whether you obtained consent from parents or guardians. If the need for consent was waived by the ethics committee, please include this information.

Response: An informed written consent was obtained from the participants before the interview was conducted. Appropriate changes have been made in the revised manuscript.

No minors were included in the study.

No retrospective data was used in the current study

Comment: When completing the data availability statement of the submission form, you indicated that you will make your data available on acceptance. We strongly recommend all authors decide on a data sharing plan before acceptance, as the process can be lengthy and hold up publication timelines. Please note that, though access restrictions are acceptable now, your entire data will need to be made freely accessible if your manuscript is accepted for publication. This policy applies to all data except where public deposition would breach compliance with the protocol approved by your research ethics board. If you are unable to adhere to our open data policy, please kindly revise your statement to explain your reasoning and we will seek the editor's input on an exemption. Please be assured that, once you have provided your new statement, the assessment of your exemption will not hold up the peer review process

Response: All data supporting this study has been included in the submission system. The statement indicating that the data will be made available upon request has been corrected in the revised manuscript.

Response to editor’s comments

Comment: The reviewers raise some important points regarding ensuring clarity in the methods and presentation of the findings. It would also be good to expand on the implications of your findings for health policy and practice. How can these fit into the wider context of reproductive health issues in a low-income setting?

Response: Thank you for your insightful comment. The appropriate changes have been made in the revised manuscript.

Reviewer #1

Comment: 1. In the abstract section (line 24-25) the authors wrote “A semi-structured interview approach was used to elicit qualitative responses from 18 fertility clients who were seeking treatment for infertility at four fertility clinics in Ghana”. The authors should clarify this because it is unclear whether the study was among those seeking infertility treatment or those who were unsuccessful with their treatment, as indicated in line 19-21 in the abstract session.

Response: The current study formed part of a broader study where couples seeking fertility treatment were initially enrolled to assess the outcome. Within this period, those who were unable to conceive after over 12 months of follow up were recruited and interviewed. These explanations were clearly indicated in the methods (participants selection section) of the study.

Comment #2: At the conclusion session of the abstract, the authors wrote “Relational activities helped”. How does it help? The authors should clarify this.

Response: Necessary changes have been made in the revised manuscript.

Comment #3: In the introduction section (line 53-55), the authors wrote “Nonetheless, research is yet to explore the coping strategies adopted by Ghanaian couples to manage psychological distress related to unsuccessful infertility treatment to inform the provision of holistic care”. I don’t think this is the case. A recent study by Dialo et al., (2024) “Exploring the psycho-social burden of infertility: Perspectives of infertile couples in Cape Coast, Ghana” provided valuable insight into this phenomenon. The current study should be a build-up on the previous one. My understanding is that the current study was conducted in the same setting. So, the authors should highlight coping strategies after unsuccessful treatment and focus on that. This distinction should be made clearer.

Response: Authors appreciate the insightful observations from the reviewer. We would like to emphasize that our study focused on patients who received treatment but had unsuccessful treatment outcomes after over 12 months of follow-up. The study by Diallo et al (2024) and a few others, assessed the coping strategies of Ghanaian couples seeking fertility care. Such studies did not focus on couples who had experienced infertility treatment failures, which presupposes that those studies were probably conducted before initiation of treatment, or during treatment. Furthermore, while we acknowledge the impressive work by Diallo et al (2024), our work is distinct from theirs, considering the multicentric approach of our study and the timing of data collection. We collected data from four fertility centers in the Cape Coast metropolis whiles Dialo et al focused on only one center.

Comment #4: At the “Materials and Methods” section (Line 130 -131) the authors wrote “Permission was sought from individual respondents before the interviews were audio-recorded”. I suggest the authors change the word “respondent” to “participants” to be consistent with the qualitative method of inquiry adopted by the authors. Please, go through the manuscript and correct this accordingly

Response: Thank you for your observation. Appropriate changes have been made in the revised manuscript

Comment #5: With regards to conducting the interviews, did the interviewer interview the participants as couples or individually? If individually, how were the female participants matched with the male participants? If they were interviewed as couples, what is the justification? The authors need to clarify these issues. My understanding from the limitation session is that the interviews were conducted individually. So, this needs to be made clear in the write-up.

Response:

The interviews were conducted on an individual basis. In typical fertility clinic settings in Ghana, it is common for one partner—most often the woman—to initially seek treatment before the male partner is invited by the specialist. This pattern generally continues during follow-up visits. Cultural perceptions that attribute infertility primarily to women often lead to men being less willing to participate in the treatment process.

As such, during follow-up visits when the interviews were conducted, participants were approached individually, based on their presence at the clinic, and only after obtaining their consent. None of the interviewees were part of a couple pair (as illustrated in Figure 1), and therefore no matching between partners was undertaken. Each participant's responses were treated as independent individual perspectives.

Due to the greater likelihood of women attending follow-up visits, a higher number of female participants were interviewed compared to male participants.

The manuscript has been revised accordingly

Comment #6: Line 136-137, the authors wrote “The interviews that were conducted in Twi were transcribed verbatim by the principal investigators”. What mechanisms were put in place to ensure that meaning is not lost during the transcription and translation processes? The authors need to provide readers with that information.

Response: Appropriate changes have been made in the revised manuscript.

Comment #7: With regards to the data analyses, did the authors employ any qualitative data analyses software? My understanding is that the analyses were done manually. This should also be made clear in the write-up.

Response: Thank you for your observations and input. Appropriate corrections have been made in the revised manuscript

Comment #8: In table 2: Themes and sub-themes that emerged from the interview, Category: 2 Coping with Uncertainty, the sub-theme under “Divine Support” “2. Pursue treatment”. Can this sub-theme be clarified? What kind of treatment? Spiritual treatment? Spiritual-herbal treatment??? Or what? The authors need to clarify this statement.

Response: Thank you for your observations: The phrase “pursue treatment” was in the context of seeking further medical treatment. Appropriate changes have been made in the revised manuscript.

Comment #9: Some of the quotes do not have unique identifiers. The authors should read through the entire manuscript and ensure that all the quotes have unique IDs.

Response: Thank you for your observation: All IDs relating to the quotes were stated, either in the preambles before the quotes, or were indicated right after the quotes. After going through the entire work, all IDs relating to various quotes were clearly indicated in the write up.

Comment #10: Can the authors replace the concept “social distancing” with “self-imposed social isolation” or “social disengagement”? Social distancing in this context could be misleading.

Response: suggested changes have been made in the revised manuscript

Reviewer #2

Comment: The methodology seems well-structured and relevant for the research objectives, particularly in addressing emotional, psychological, and social coping mechanisms. However, refining data collection techniques, and enhancing discussion of data validity and ethical considerations could strengthen the study’s rigor and impact. Further comments in the manuscript.

Response: The methodology has been revised to enhance the study’s rigor

Comment: Consider rewording for clarity. I recommend that you send your manuscript for editing as there are some grammatical issues and inconsistencies throughout your manuscript. (Line 110-111)

Response: The suggested amendments have been made in the revised manuscript.

Comment: How was trustworthiness ensured in this study? There isn’t mention. Ensuring trustworthiness is crucial in qualitative research given its subjective nature.

Response: The suggested corrections have been made in the revised manuscript

Comment: Confirmation if there were more than one participant investigators

Response: Yes, the interviews were conducted by more than one principal investigator but at different locations/study sites.

Comment: The manuscript mentions Thematic Content Analysis (TCA) and cites Braun & Clarke (19). Their framework is for Thematic Analysis (TA), kindly check and align. Additionally, ensure that reference (19) is correctly formatted in the reference list.

Response:

Thank you for drawing our attention to this important distinction. You are correct that Braun and Clarke’s framework refers specifically to Thematic Analysis (TA) and not Thematic Content Analysis (TCA). We have reviewed the manuscript and made the necessary corrections to ensure consistent and accurate terminology throughout. All references to "Thematic Content Analysis" have been revised to "Thematic Analysis" in line with Braun and Clarke’s methodology.

Additionally, we have verified and corrected the formatting of reference (19) in the reference list to ensure it conforms to the journal’s style requirements.

We appreciate your careful review and helpful suggestion.

Comment: Given the sensitive nature of infertility, This study did not incorporate emotional support for participants or referrals to counselling if needed.

How was confidentiality maintained, especially in small community settings where participants might be known to each other?

Response: Thank you for this important observation. We acknowledge that the study did not formally incorporate mechanisms for emotional support or referrals to counseling services. We recognize the ethical importance of providing psychosocial support, especially in sensitive topics such as infertility. We have therefore included a statement in the limitations section in the revised manuscript to recommend that future studies integrate structured emotional support or referral systems as part of ethical safeguards

Additionally, Given the sensitive nature of the study, extra precautions were taken to protect participants’ confidentiality. This included assigning special codes to mask identifiable details, and securing collected data in a password protected computer.

Comment: Some references are incomplete or missing proper formatting. For example, "According to [24]" should specify the author's name rather than just the citation number.

Ensure all in-text citations have corresponding references in the reference list.

Ensure consistent use of “couples” vs. “individuals” when referring to coping mechanisms.

The discussion is well-researched and informative, but improving clarity, and ensuring consistency in terminology will enhance readability

The study seems to have focused on individual perspectives rather than dynamic couples. Including both partners in discussions could provide deeper relational insights.

Response:

Thank you for your insightful comment. You are absolutely right that including both partners in discussions could offer richer relational insights into the dynamics of infertility experiences and treatment-seeking behaviors. While this study focused on individual perspectives due to the practical realities of clinic attendance—where partners often presented at different times—we acknowledge the value of capturing shared experiences and interactions within couples.

Future research could build on these findings by adopting a dyadic approach, incorporating joint interviews or matched partner perspectives to explore how couples navigate infertility together. This would undoubtedly deepen our understanding of the relational context and shared decision-making processes involved. We have, however, revised the discussion section to reflect the significance of engaging the couple pair in the study

Other corrections have been effected in the revised manuscript.

Comment: The limitations briefly mention low male participation, but it could be expanded for instance:

-How did this impact the findings?

-Could this limitation have introduced gender bias?

-What recommendations do you have for future research?

- You mentioned low male participation as a limitation. Consider discussing how this affects the transferability of findings and whether future studies should use targeted recruitment strategies (e.g., male-focused support groups, incentives?).

Response:

Thank you for your constructive feedback. We appreciate your suggestions and have revised the limitations section to provide a more comprehensive discussion of the impact of low male participation on our study findings. Please find our response below:

We acknowledge that the limited participation of male partners may have influenced the findings by providing a more female-centric perspective on infertility experiences and treatment-seeking behaviors. This imbalance could introduce gender bias, particularly in understanding the shared experiences, emotional responses, and decision-making processes within couples. As such, the transferability of the findings to all individuals experiencing infertility—particularly men—may be constrained.

To address this, we have expanded the discussion section to acknowledge how this limitation may affect the interpretation of results and the generalizability of the findings. Additionally, we now recommend that future studies consider targeted recruitment strategies to ensure more balanced gender representation. These could include engaging male-focused support groups, con

---

## [Decision Letter · Decision Letter 1]

Coping strategies of Ghanaian couples after unsuccessful infertility treatment: An exploratory qualitative study

PONE-D-25-06994R1

Dear Dr. Arhin,

We’re pleased to inform you that your manuscript has been judged scientifically suitable for publication and will be formally accepted for publication once it meets all outstanding technical requirements.

Kind regards,

Lily Kpobi, Ph.D.

Academic Editor

PLOS ONE

Additional Editor Comments (optional):

Please take note of the additional minor correction from Reviewer 1 and effect the necessary changes for the final manuscript.

Reviewers' comments:

Reviewer's Responses to Questions

**Comments to the Author**

1. If the authors have adequately addressed your comments raised in a previous round of review and you feel that this manuscript is now acceptable for publication, you may indicate that here to bypass the “Comments to the Author” section, enter your conflict of interest statement in the “Confidential to Editor” section, and submit your "Accept" recommendation.

Reviewer #1: (No Response)

Reviewer #2: All comments have been addressed

2. Is the manuscript technically sound, and do the data support the conclusions?

Reviewer #1: Yes

Reviewer #2: Yes

3. Has the statistical analysis been performed appropriately and rigorously? 

Reviewer #1: N/A

Reviewer #2: Yes

4. Have the authors made all data underlying the findings in their manuscript fully available?

Reviewer #1: Yes

Reviewer #2: (No Response)

5. Is the manuscript presented in an intelligible fashion and written in standard English?

Reviewer #1: Yes

Reviewer #2: Yes

6. Review Comments to the Author

Reviewer #1: Reviewer Reaction: The response to the first comment is unsatisfactory. The information in the abstract should be consistent with the information in the paper. The justification is the authors gave is that those who were unable to conceive after 12 months of follow up were deemed as unsuccessful with the treatment. I insist that the authors should make this correction in the abstract to reflect this fact in the paper. This is particularly important following the response the authors provided to comment #3. So, I suggest the statement should read as “A semi-structured interview approach was used to elicit qualitative responses from 18 fertility clients after unsuccessful treatment at four fertility clinics in Ghana”.

Reviewer #2: The comments made in the previous review was addressed to my satisfaction. Nothing to attach. I want to make note that the question above on data availability I couldn't respond as I am unaware if authors have made the study data available

7. PLOS authors have the option to publish the peer review history of their article (what does this mean? ). If published, this will include your full peer review and any attached files.

**Do you want your identity to be public for this peer review?** For information about this choice, including consent withdrawal, please see our Privacy Policy .

Reviewer #1: No

Reviewer #2: **Yes: ** Hlolisile Chiya

---

## [Editor Report · Acceptance letter]

PONE-D-25-06994R1

PLOS ONE

Dear Dr. Arhin,

I'm pleased to inform you that your manuscript has been deemed suitable for publication in PLOS ONE. Congratulations! Your manuscript is now being handed over to our production team.

Kind regards,

on behalf of

Dr. Lily Kpobi

Academic Editor

PLOS ONE